# Benefits of Playing at School: Filler Board Games Improve Visuospatial Memory and Mathematical Skills

**DOI:** 10.3390/brainsci14070642

**Published:** 2024-06-26

**Authors:** Verónica Estrada-Plana, Andrea Martínez-Escribano, Agnès Ros-Morente, Maria Mayoral, Agueda Castro-Quintas, Nuria Vita-Barrull, Núria Terés-Lleida, Jaume March-Llanes, Ares Badia-Bafalluy, Jorge Moya-Higueras

**Affiliations:** 1Department of Psychology, Sociology and Social Work, University of Lleida, 25001 Lleida, Spain; veronica.estrada@udl.cat (V.E.-P.); agnes.ros@udl.cat (A.R.-M.); nuria.vita@udl.cat (N.V.-B.); ntlleida@gmail.com (N.T.-L.); jaume.march@udl.cat (J.M.-L.); badiaares@gmail.com (A.B.-B.); 2Parc Sanitari de Sant Joan de Déu de Sant Boi, C/del Antoni Pujadas, 42, 08830 Sant Boi de Llobregat, Spain; 3Instituto de Investigación Sanitaria Gregorio Marañón (IISGM), 28009 Madrid, Spain; maria.mayoral@salud.madrid.org; 4Centre for Biomedical Research Network on Mental Health (CIBERSAM), Instituto de Salud Carlos III, 28029 Madrid, Spain; aguedacastro@ub.edu; 5Faculty of Biology, University of Barcelona, 08007 Barcelona, Spain

**Keywords:** cognitive intervention, game-based intervention, primary education, working memory, maths, modern board games

## Abstract

The aim of the study was to test the effectiveness of cognitive interventions based on modern board games in school settings to improve memory outcomes and math skills. A parallel, quasi-experimental study was carried out with children (*n* = 234) into third and fourth grades (8–10 years old). School centres were allocated into a general domain intervention (playing memory board games), a specific domain intervention (playing mathematical board games) or a control group (regular classes without playing). Teachers carried out bi-weekly sessions during the last 30 min of mathematical lessons (8 weeks, 15 sessions). Before and after intervention, we individually measured verbal and visuospatial memory outcomes (short-term memory and working memory updating) and mathematical skills (number operations, number ranking, number production and problem solving). The results showed significant transfer effects of both memory and math trainings. In third grade, we found that playing math games showed medium–large effect sizes in visuospatial short-term memory and updating memory, number operations and number ranking compared to the control group. In fourth grade, we found that playing memory games showed significant small effect sizes in problem solving compared to the control group. Playing board games could be a methodology that enhances cognitive and mathematical development in children.

## 1. Introduction

Working memory (WM) is a fundamental cognitive process in scholastic skills and achievement [1]. WM is defined as the interaction of cognitive processes that are involved in keeping information accessible and manipulating it while performing complex tasks, such as learning, reasoning or comprehension [2]. The multicomponent WM model [2] is the model that better fit for school age-children [3]. This model has been changing over time [4]. The concept of WM evolved from the short-term memory (STM) concept [4], which is understood as the storage of information for brief periods of time [2]. Thus, the first model of WM included two temporary components: the phonological loop which is related to the storage of verbal information and the visuospatial sketchpad which is related to the storage of visuospatial information [2,5]. In addition, the model included the central executive which is related to the attentional control system and the manipulation of information [2,5]. Parallel studies in executive functions (EFs) included some tasks that require the monitoring and updating of WM information [6]. This function allows the WM to modify content, considering its relevance, and to replace the old information to accommodate new inputs. In fact, executive-loaded WM tasks can be divided into complex WM tasks and updating tasks [7]. Previous studies in school-age children show that both tasks belong to the WM construct, but only updating tasks are related to EFs [8]. 

Previous studies show that STM and WM updating processes could be relevant outcomes for mathematics in preschool and school ages [9,10]. Thus, improving STM and WM updating may be essential in the acquisition of mathematical concepts in school-age children. 

The main strategies to increase STM and WM updating are called cognitive-focused interventions [11]. One of these interventions is called cognitive training (CT), which “*entails repeated exercise of a specific cognitive process over a period to improve performance on the trained task as well as on tasks that were not specifically trained*” [12]. Besides improvements in WM tasks [13], WM training has been related to increases in the activity of brain regions linked to WM, such as the frontoparietal and temporal regions [14]. Following the definition of CT, improvements can take place on tasks that may not be specifically trained, which are called transfer effects [15]. Some authors [15,16] have suggested that there are different kinds of transfer effects. On the one hand, near transfer consists in an improvement in the same trained domain but in a different task than the trained one. On the other hand, far transfer consists in an improvement in a task that requires a different but related cognitive construct from the trained one or a completely different task that depends on the trained cognitive processes. Different meta-analyses concluded that near transfer effects in STM and WM tasks are possible after a WM training in school-age children [17,18,19]. However, empirical and meta-analysis studies are not consistent in the conclusion about far transfer effects on mathematical skills. For instance, some studies found improvements in mathematical tasks after having been trained with a WM cognitive training [20], whereas other studies did not find significant results in mathematical skills [17]. In addition, other authors [21] differentiate general and specific domain trainings. General domain trainings consist in training cognitive processes that play a critical role in mathematical development, as would be the case of the STM and WM trainings listed above. On the other hand, specific domain trainings are those that train specific skills, such as mathematical tasks. Previous studies found that mathematical interventions can improve mathematical skills, but also WM measures [21,22]. This suggests a bidirectional relation between WM and mathematical skills [23].

All of the studies cited above included game elements in computerized cognitive trainings. Game-based computerized CTs have found significant results in WM tasks [24,25]. Moreover, these studies have found more improvements in WM CTs with game elements than in the same CTs without them. However, these interventions did not include social interaction, which could have a positive impact on cognition [26]. Hence, it is important to include elements that not only boost cognitive tasks but also social components that could impact educative outcomes.

Recently, board games have been receiving more attention in research [27]. They are defined as games with a board with pieces (and/or cards) on it, with predefined rules that fix the number of pieces/cards, the number of positions of the elements, and the number of their possible moves [28]. Board games can be classified into traditional, mass-market or modern board games [29]. On the one hand, traditional and classical games are those that do not have a known author and do not have commercial rights (i.e., chess). On the other hand, mass market games are those that are commercialized massively but without a known author or innovative mechanics, like puzzles. Finally, modern board games are edited with an attractive and visual appearance, with different mechanics and originally designed by a known author or company [29]. In addition, modern board games can be commercialized in an extended way with relatively economical prices.

This classification is useful to classify and identify board games considering their timeline of creation. However, they can also be classified considering their simplicity and length. Filler board games are not explicitly included in the classification of board games [29]. The main feature of filler board games is that they are set up quickly and last between 15 to 20 min; they are not filled to exceed 30 min. Thus, filler games are brief, can have several but simple mechanics and can be learned quickly because of their simple rules [30]. These characteristics may make these board games a good option to be included in trainings.

Previous research was focused on computerized CTs. However, interventions based on board games could have the potential to be beneficial to cognitive processes and mathematical skills. In school-aged children, Jirout & Newcombe [31] found a relation between block-play, puzzles and board games and higher spatial abilities. Newman et al. [32] found a relation between the use of block-play and mental rotation. Furthermore, Nath & Szücs [33] found that level in construction games is related to visuospatial WM and this is related to mathematical performance.

As far as we know, there are only a few studies that focused on testing possible improvements on WM updating using brief ad hoc board games or modern board games [34,35,36]. In addition, some studies found improvements in STM after training this cognitive process with brief ad hoc board games [37,38] or after playing the traditional Go game [39]. All of the studies mentioned above showed improvements in STM or WM updating after playing board games. One of them found a significant transfer effect on a mathematical measure [38]. Also, some studies found that training mathematical skills with specific tasks produces near transfer effects in preschool ages [40,41,42,43]. Regarding preschool ages, two studies compared a STM and WM general domain intervention and a mathematical specific intervention with brief board games created ad hoc [44,45]. Passolunghi & Costa [45] found that an intervention focused on STM and WM improved complex WM tasks and early numeracy skills. However, Scalise et al. [44] did not find an improvement on visuospatial STM after playing a memory card game. However, they found an improvement on early numeracy skills after playing a memory card game and especially after playing a mathematical card game. Most of the studies used brief board games, but no study was developed in school-age children for training STM, WM updating and mathematical skills through specific and general domain interventions.

To conclude, the aim of this study was to assess the effectiveness of general domain and specific trainings based on filler board games in a school setting in children between 8 and 10 years old, following CONSORT guidelines adapted to non-randomized trials [46]. 

Considering near transfer effects, we hypothesized that the following: (i) those children who were trained at playing filler memory board games would improve their STM and WM updating abilities after the intervention more than the mathematical group and the control group; (ii) those children who were trained at playing filler mathematical board games would improve their mathematical skills after the intervention more than the memory group and the control group. Considering far transfer effects, (i) those children who were trained at playing memory board games would improve their mathematical skills after the intervention more than the control group; (ii) those children who were trained at playing mathematical board games would improve their STM and WM updating abilities after the intervention more than the control group.

## 2. Materials and Methods

### 2.1. Participants

The needed sample size was calculated considering the results obtained in verbal STM from Estrada-Plana et al. [35]. Sample size calculation was performed for a two-sided hypothesis test, 90% statistical power and an alpha level of 5%. Following this study [35], we also considered a 50% risk of possible losses. The analysis revealed that 11 subjects per group were enough for the present research. Initially, 343 children were contacted (see Figure 1). Inclusion criteria were the following: (i) studying in the second grade—3rd and 4th—of primary school, (ii) studying in one of the 7 different schools of both rural and urban areas of Lleida (Catalonia, Spain) selected by the Department of Education to participate in the present research, and (iii) having signed the informed consent. Exclusion criteria were the following: (i) having participated in similar previous research, (ii) showing difficulties with the Spanish or Catalan languages, (iii) having a lack of basic sociodemographic information, (iv) missing data in some primary or secondary outcomes. After matching the inclusion criteria, 270 children were allocated into experimental or control groups. Next, we excluded some of them for the following reasons (see Figure 1): (i) eight children had participated before in a previous pilot study of our research group; (ii) five children whose date of birth and one child whose school-grade level were not collected during the assessment period; (iii) two children dropped out of their school after the intervention sessions started; (iv) seven children did not carry out the assessment individually at pre or post times for other reasons different from dropping out of schools (i.e., being ill); (v) one child did not carry out the assessment in a group session; (vi) seven children did not carry out the individual or group assessment at some time during the study; and (vii) five parents did not complete rating scales about their son or their daughter. Finally, taking the data above into account, the final sample consisted of 234 children. Participants who were allocated into intervention or control groups had a higher socio-economic level than those children who were not included in the analysis (*U* = 1979.50, *p* = 0.037, *r* = 0.26). In addition, participants who were allocated into intervention or control groups had higher verbal STM than those children who were not included in the analysis (*U_Verbal STM(hits)_* = 2784.00, *p* = 0.010, *r* = 0.27). Finally, participants who were allocated into intervention or control groups had higher mathematical skills than those children who were not included in the analysis (*U_Number Operations_* = 2966.00, *p* < 0.001, *r* = 0.37; *U_Number Ranking_* = 3813.00, *p* = 0.002, *r* = 0.29; *U_Number Production_* = 3067.00, *p* < 0.001, *r* = 0.34; *U_Problem Solving_* = 3016.50, *p* < 0.001, *r* = 0.36). 

In third grade, students begin to calculate additions and subtractions, while in fourth grade they are supposed to master both operations. To control the effects of development and mastery level because of the grade, the sample was divided considering elementary primary school grades [3]. Subsample 1 consisted of children who were in third grade (*n* = 121) and subsample 2 consisted of children who were in fourth grade (*n* = 113). Demographical and psychological data about children in the 3rd and 4th grades subsamples are summarized in Table 1. Participants who repeat a course ($\chi^{2}$*3rd* (1) = 1.25, *p* = 0.870, *V* = 0.07; $\chi^{2}$*4th* (1) = 1.45, *p* = 0.484, *V* = 0.11) and who had a psychological diagnosis ($\chi^{2}$*3rd* (1) = 1.25, *p* = 0.535, *V* = 0.10; $\chi^{2}$*4th* (1) = 3.73, *p* = 0.155, *V* = 0.18) were equally distributed in the groups.

### 2.2. Measures

#### 2.2.1. Covariate Measures

Previous studies showed that age [3], socio-economic status (SES) [47], fluid reasoning [48,49] and math anxiety [50] could be related to STM and WM or could modulate cognitive trainings. Therefore, the following covariates were assessed in the present study: age in years, SES index (the formula’s index was: ([education scale score] × 3) + ([occupation scale score] × 5)) [51] and fluid reasoning assessed by Raven’s Progressive Matrices Test (RPMT) [52]. Finally, mathematical anxiety was assessed by the Abbreviated Math Anxiety Scale (AMAS) [53]. This outcome was only considered in the analyses of mathematical competences.

#### 2.2.2. Memory Outcome Measures

*Verbal STM*. We used the forward Digits test from WISC-IV [54] to assess verbal STM. The participant must exactly repeat the digit sequence that the researcher conveys. The difficulty of the task increases gradually (from two to nine digits). The task finishes when participants make a mistake in two trials of the same difficulty. For every sequence remembered, one point was given. The measure included in the study was the sum of the trials repeated correctly. The task lasted 5 min.

*Visuospatial STM*. We used the forward Corsi block test to assess visuospatial STM following the procedure of Andersson & Lyxell [55]. The child was instructed to observe the experimenter tap a sequence of blocks for a rate of one per second and then to attempt to repeat the sequence in the same order. Every span size had two different sequences of two to nine blocks. The task finished when participants made a mistake in two sequences of the same difficulty. For every sequence remembered, one point was given. The measure included in the study was the sum of the trials repeated correctly. The task lasted 5 min.

*Verbal WM Updating*. The Keep Track Task was administered according to the guidelines presented in Tamnes et al. [56]. Participants saw different semantic categories (animals, clothing, colours, countries, fruits and relatives) on the computer. Below each category, 3 words could appear related to its correspondent category (18 words in total). Each word appeared in a pseudorandomized order with a ratio of 2000 ms. The task itself consisted of two practice trials with two and three categories, four trials with three categories, four trials with four categories, and one with five categories. Subjects had to recall the last word presented in each one of the target categories. The outcome assessor registered each response. The task ended when all of the trials were administered. The measure included in the study was the sum of the correct trials (maximum 33 words to be recalled). The duration of the task was 10 min.

*Visuospatial WM Updating*. The Keep Track Task was created following the guidelines from Tamnes et al. [56] in order to assess the visuospatial component of updating. A matrix of 3 × 3 was shown on the computer screen for each trial. The targets consisted of 6 different faces from different colours (white, black, yellow, green, red and blue). The faces appeared on the screen, in a changeable number of presentations (between one and five). Each face appeared in a pseudorandomized order with a ratio of 2000 ms. The task itself consisted of two practice trials with two and three colours, four trials with three different colours, four trials with four different colours, and one trial with five different colours. The task ended when all of the trials were administered. The aim was to recall the last position of the faces. The outcome assessor registered each response. The measure included in the study was the sum of the correct trials. The duration of the task was 10 min.

#### 2.2.3. Mathematical Outcome Measures

Alloway & Passolunghi [57] considered different tasks to assess mathematical skills (the first three tasks below). Hence, we adapt the same methodology in the present study. In addition, we added a task to assess problem-solving skills. The scoring for all items was 0 (incorrect) and 1 (correct).

*Number operations task*. This task consisted of performing up to 40 mathematical written operations (20 addition operations and 20 subtraction operations) with increasing difficulty (single-digit operations, double-digit operations and multi-digit operations) within 2 min. The measure included in the study was the sum of the correct operations (maximum score = 40). Based on our sample, this measure had a good reliability (Cronbach’s α = 0.86).

*Number ranking task*. Children were prompted to order numbers from the smallest to the highest (e.g., 1-9-3-5 = 1-3-5-9). The students had a total of 16 items with increasing difficulty (from single digit to three digit numbers) and a maximum of 2 min to complete the task. The measure included in the study was the sum of the correct numbers (maximum score = 81). Based on our sample, this measure had a good reliability (Cronbach’s α = 0.85).

*Number production task*. In this task the child translated numbers from the written representation to digit number representation in units (e.g., 3 dozen = 36 units). There were 23 items and the time limit to perform the task was 2 min. The measure included in the study was the sum of the correct answers (maximum score = 23). Based on our sample, this measure had a good reliability (Cronbach’s α = 0.88).

*Problem-solving task*. The task consisted of 16 mathematical problems with increasing difficulty. The child needed to read each problem, to decide which operation to apply (addition or subtraction) and to extract the information needed to conduct the operation. In 5 min, participants were prompted to solve the highest number of problems as possible. As for the other tasks, the dependent variable used in the study was the sum of the correct problems solved (maximum score = 16). Based on our sample, this measure had a good reliability (Cronbach’s α = 0.82).

### 2.3. Procedure

Firstly, the Department of Education approved the project. Ethical requirements following the Declaration of Helsinki were approved by the University. The Department of Education selected five urban public schools and two rural public schools by a convenience sampling method. For this reason, this study was a multicentre study. Headmasters from all of the schools accepted to participate in the research. Afterward, six schools were assigned to both groups of games (Math GTG and Memory GTG). Allocation was made by a code considering similar school characteristics (ratio 1:1). Both Math GTG and Memory GTG consisted of two urban public schools and one rural public school. The last author was the person who generated the allocation sequence, who enrolled participants and who assigned participants into game groups. The Control Group (CG) was assigned by the Department of Education. This group was an urban school from higher socio-economic backgrounds. In addition, as can be seen in Table 1, performances in number operations and some memory and mathematical outcomes were higher in the CG than in the schools that played. Then, informed consent was given to the families of the children. The baseline assessment started after receiving parental authorization. Parents were asked about their social status and socio-demographic information about them and their children (e.g., age, sex and place of birth). STM, WM updating and math anxiety tests were applied individually, while mathematical skill tasks and a fluid reasoning test were assessed in groups. All of the teachers were trained in one session to apply each game systematically and to register the incidences/attendance. The handbook with the description of the intervention was delivered to teachers. The material required for the intervention (i.e., board games and the document to register incidents) was also delivered to schools. After the intervention, memory and math outcomes were assessed again in the three groups. In all of the assessments, the order of the neuropsychological tests in individual sessions was counterbalanced across participants to control assessment bias. Two psychologists, a senior research psychologist and other research assistants performed all of the baseline and post assessments. After the post-test assessment, the CG received the same intervention as Math GTG for ethical reasons. See Figure 2 for a detailed description of timeline.

### 2.4. Intervention

The intervention used in the memory group included five different board games oriented and designed to boost STM and WM updating: Alles Kanone! [58], Alles Tomate! [59], Spooky Stairs [60], Out of Mine! [61] and Chicken, Cha Cha Cha! [62]. The intervention used in the mathematical group included five different board games oriented and designed to boost mathematical skills: 7ate9 [63], Numenko in a bag [64], Pig 10 [65], Shut the box [66] and Auf zack! [67] (see a brief description of the games in Appendix A: Brief description of the games).

All of the participants were exposed to 15 sessions (2 sessions per week, 30 min each) for eight weeks. Each participant played 3 sessions of each board game (90 min). The groups of play were invariant and composed of 4 children. The intervention sessions were applied from February to April in the year 2015, although each school adapted them to their specific calendar. The participants played the board games described above as part of the math lessons. Teachers conducted the intervention sessions. Each session was described in a handbook for the teachers to systematically conduct the intervention. In addition, every teacher had a document to register incidents during the playing time (i.e., the rules of each game were not properly followed) and the assistance of the participants. All of the teachers from the Memory and the Math Game Training Groups (GTGs) knew the aim of the study, but did not know the specific hypothesis. Hence, children and teachers from both groups were masked. However, the CG was aware of its condition.

### 2.5. Statistical Analysis

All of the analyses were run with Jamovi 1.6.15 version [68]. Firstly, we calculated baseline differences in the main outcomes and demographic/psychological characteristics. Considering that the mean scores are generally not normally distributed, we used the Kruskal–Wallis test to compare quantitative outcomes. Dwass–Steel–Critchlow–Fligner post-hoc pairwise comparisons were calculated to identify differences between groups. We used chi-square to compare nominal variables, such as sex or ethnicity. Epsilon squared (ε^2^) and Cramer’s V were calculated as effect size measures [69], respectively. Secondly, changes before and after the intervention were tested between experimental and control conditions. A mixed-model analysis of covariance (ANCOVA) was used to calculate time by group interaction effects for each dependent variable. Group (Math GTG, Memory GTG and CG) was a fixed factor and time was a repeated measures factor (pre and post). We included subjects as random effects. The centred covariates included in the model were age, SES index, fluid intelligence (RPMT score) and math anxiety (AMAS score). Post-hoc pairwise comparisons adjusted with Bonferroni correction were calculated to identify differences between groups. We calculated simple main effects and compared the magnitude of the effects using Cohen’s d. We followed the formula of Brysbaert & Stevens [70] to calculate Cohen’s d. We interpreted Cohen’s d as follows [69]: d < 0.20 = trivial; 0.20 ≤ d < 0.50 = small; 0.50 ≤ d < 0.80 = medium; and d ≥ 0.80 = large. We calculated Cohen’s d between the Math GTG and the CG and between the Math GTG and the Memory GTG.

## 3. Results

### 3.1. Compliance with the Intervention Program

In Grade 3, all of the participants received between 13 and 15 sessions (98.84% attendance in the Math GTG; 100% attendance in the Memory GTG), while in Grade 4, all of the participants received between 11 and 15 sessions (98.82% attendance in the Math GTG; 99.09% attendance in the Memory GTG).

### 3.2. Descriptive and Baseline Comparisons

#### 3.2.1. Grade 3

First of all, we found significant differences in math anxiety (χ^2^ (2) = 11.11, *p* = 0.004, *ε*^2^ = 0.09). Participants from the Math GTG had higher math anxiety than the Memory GTG (*W* = −3.87, *p* = 0.017) and CG (*W* = 3.57, *p* = 0.031). Although we found a significant result for age (*χ*^2^ (2) = 7.13, *p* = 0.028, *ε*^2^ = 0.06), we did not find any significant difference in pairwise comparisons. In addition, we found significant differences in number operations skills (*χ*^2^ (2) = 17.77, *p* < 0.001, *ε*^2^ = 0.15), number ranking skills (*χ*^2^ (2) = 30.17, *p* ≤ 0.001, *ε*^2^ = 0.25) and problem-solving skills (*χ*^2^ (2) = 19.52, *p* ≤ 0.001, *ε*^2^ = 0.16). Participants from the Math GTG had lower scores than the CG in these measures (*W_Number Operations_* = −5.80, *p* ≤ 0.001; *W_Number Ranking_* = −7.60, *p* < 0.001; *W_Problem Solving_* = −5.98, *p* < 0.001). Finally, participants from the Memory GTG had higher scores than the Math GTG in number ranking (*W* = 3.75, *p* = 0.022) and lower scores than the CG in number operations (*W* = −3.32, *p* = 0.049) and problem solving (*W* = −4.56, *p* = 0.004).

#### 3.2.2. Grade 4

First of all, we found significant differences in SES index (*χ*^2^ (2) = 8.92, *p* = 0.012, *ε*^2^ = 0.08). Participants from the Math GTG and from the Memory GTG had a lower SES index than the CG (*W_Math GTG_* = −3.42, *p* = 0.042; *W_Memory GTG_* = −3.44, *p* = 0.040). In addition, we found significant differences in verbal WM (*χ*^2^ (2) = 9.47, *p* = 0.009, *ε*^2^ = 0.09), in visuospatial STM (*χ*^2^ (2) = 7.47, *p* = 0.024, *ε*^2^ = 0.07) and in visuospatial WM (*χ*^2^ (2) = 11.25, *p* = 0.004, *ε*^2^ = 0.10). Participants from the Memory GTG had lower scores than the CG in visuospatial STM (*W* = −3.71, *p* = 0.024). Participants from the Math GTG and from the Memory GTG had lower scores than the CG in verbal WM (*W_Math GTG_* = −3.70, *p* = 0.024; *W_Memory GTG_* = −3.74, *p* = 0.022) and visuospatial WM (*W_Math GTG_* = −3.40, *p* = 0.043; *W_Memory GTG_* = −4.47, *p* = 0.004). Finally, we found significant differences in number operations skills (*χ*^2^ (2) = 11.44, *p* = 0.003, *ε*^2^ = 0.10). Participants from the Memory GTG had lower scores than the CG in this measure (*W* = −4.75, *p* = 0.002).

### 3.3. Intervention Effects in Grade 3

#### 3.3.1. Memory Outcomes

We found significant differences between pre and post scores in all of the memory outcomes, except for verbal STM (from *F*_Visuospatial STM(span)_ (1,116) = 4.66, *p* = 0.033, *d* = 0.25 to *F*_Visuospatial WM_ (1,117) = 13.72, *p* ≤ 0.001, *d* = 0.39; see all the results in Table 2). Significant main interactions were found between groups and times in visuospatial STM (*F*_Hits_ (2,116) = 3.32, *p* = 0.040; *F*_Span_ (2,116) = 5.19, *p* = 0.007) and visuospatial WM updating (*F* (2,116) = 3.55, *p* = 0.032) (see Table 2). For the visuospatial STM and WM updating, simple main effects showed that scores in the Math GTG were significantly higher after the intervention (*F*_Visuospatial STM (hits)_ (1,117) = 17.36, *p* ≤ 0.001; *F*_Visuospatial STM (span)_ (1,117) = 21.89, *p* ≤ 0.001; *F*_Visuospatial WM_ (1,117) = 32.61, *p* ≤ 0.001). The simple main effect showed that scores in the Memory GTG were nearly significant after the intervention in WM updating (*F*_Visuospatial WM_ (1,116) = 3.77, *p* = 0.055), but were not significant in visuospatial STM (*F*_Visuospatial STM (hits)_ (1,117) = 1.60, *p* = 0.208; *F*_Visuospatial STM (span)_ (1,115) = 0.69, *p* = 0.408). Regarding the CG, the simple main effect showed that scores in this group did not change after the intervention (*F*_Visuospatial STM (hits)_ (1,116) = 0.20, *p* = 0.653; *F*_Visuospatial STM (span)_ (1,115) = 0.69, *p* = 0.406; *F*_Visuospatial WM_ (1,116) = 0.16, *p* = 0.690). The change in the Memory GTG in comparison to the CG showed small effect sizes in visuospatial STM (hits: *d* = 0.41; span: *d* = 0.42) and visuospatial WM updating (*d* = 0.37). However, the change in the Math GTG in comparison to the CG showed significant medium–large effect sizes in visuospatial STM (hits: *d* = 0.61; span: *d* = 0.85) and visuospatial WM updating (*d* = 0.58) (see Figure 3A–C).

#### 3.3.2. Math Skills

We found significant differences between pre and post scores in all of the mathematical outcomes (from *F_Problem Solving_* (1,121) = 8.08, *p* = 0.005, *d* = 0.38 to *F_Number Operation_* (1,116) = 61.68, *p* < 0.001, *d* = 0.83; see all of the results in Table 2). In addition, we found significant group differences in number ranking (*F* (2,117) = 7.13, *p* = 0.001) and problem solving (*F* (2,117) = 7.36, *p* ≤ 0.001). The CG had higher global scores than the Math GTG in number ranking skills (*t* (109) = 3.67, *p* = 0.001) and problem-solving skills (*t* (110) = 3.30, *p* = 0.004). The CG had higher global scores than the Memory GTG in problem-solving skills (*t* (127) = 3.33, *p* = 0.003), but not in number ranking skills (*t* (127) = 1.03, *p* = 0.914). Moreover, significant interactions were found between groups and times in number operation skills (*F* (2,113) = 3.73, *p* = 0.027) and number ranking skills (*F* (2,117) = 6.22, *p* = 0.003) (see Table 2). Simple main effects showed that scores in number operations were significantly higher after the intervention in the three groups (*F_Math GTG_* (1,108) = 124.59 *p* ≤ 0.001; *F_Memory GTG_* (1,122) = 8.08, *p* = 0.005; *F*_CG_ (1,107) = 15.47, *p* ≤ 0.001). Simple main effects showed that scores in number ranking skills were significantly higher after the intervention in the three groups (*F_Math GTG_* (1,108) = 114.34, *p* ≤ 0.001; *F_Memory GTG_* (1,132) = 7.35, *p* = 0.008; *F*_CG_ (1,106) = 6.74, *p* = 0.011). The change in the Memory GTG in comparison to the CG showed trivial and small effect sizes in number operations skills (*d* = 0.07) and number ranking skills (*d* = 0.32), respectively. However, the change in the Math GTG in comparison to the CG showed medium and large effect sizes in number operations skills (*d* = 0.48) and number ranking skills (*d* = 0.91), respectively (see Figure 3D,E).

### 3.4. Intervention Effects in Grade 4

#### 3.4.1. Memory Outcomes

We found significant differences between pre and post scores in all of the memory outcomes, except for verbal STM (from *F*_Visuospatial STM(span)_ (1,109) = 3.96, *p* = 0.049, *d* = 0.26 to *F*_Visuospatial WM_ (1,109) = 33.65, *p* ≤ 0.001, *d* = 0.68; see all of the results in Table 1 and Table 2). In addition, we found significant group differences in visuospatial STM (*F*_Hits_ (2,106) = 4.90, *p* = 0.009; *F*_Span_ (2,106) = 3.55, *p* = 0.032) and in visuospatial WM updating (*F* (2,106) = 5.10, *p* = 0.008). The CG had higher global scores than the Memory GTG in visuospatial STM (hits) (*t* (106) = 3.12, *p* = 0.007). The CG had higher global scores than the Math GTG (*t* (106) = 2.66, *p* = 0.027) and Memory GTG (*t* (106) = 2.89, *p* = 0.014) in visuospatial WM updating. Although we found main group differences in visuospatial STM span, we only found nearly significant post hoc comparisons (comparison between the CG and Math GTG: *t* (106) = 2.34, *p* = 0.063; comparison between the CG and Memory GTG: *t* (106) = 2.30, *p* = 0.070). Considering simple main effects, all of the participants improved in visuospatial WM updating (*F*_Math GTG_ (1,109) = 9.01, *p* = 0.003; *F*_Memory GTG_ (1,109) = 31.84, *p* ≤ 0.001; *F_CG_* (1,109) = 7.13, *p* = 0.009). Participants from the Math and Memory GTGs improved verbal WM updating (*F*_Math GTG_ (1,108) = 4.77, *p* = 0.031; *F*_Memory GTG_ (1,108) = 9.20, *p* = 0.003; *F*_CG_ (1,108) = 0.08, *p* = 0.605) and only the Memory GTG improved visuospatial STM (hits) (*F*_Math GTG_ (1,109) = 0.28, *p* = 0.779; F_Memory GTG_ (1,109) = 8.05, *p* = 0.005; *F*_CG_ (1,109) = 1.78, *p* = 0.185). However, no significant interactions were found (see Table 1 and Table 2).

#### 3.4.2. Math Skills

We found significant differences between pre and post scores in all of the mathematical outcomes (from *F_Problem Solving_* (1,97) = 27.40, *p* ≤ 0.001, *d* = 0.49 to *F_Number Production_* (1,100) = 60.92, *p* < 0.001, *d* = 0.72; see all of the results in Table 1 and Table 2). In addition, we found significant group differences in number operations (*F* (2,100) = 9.33, *p* ≤ 0.001) and number production (*F* (2,106) = 3.48, *p* = 0.034). The CG had higher global scores than the Memory GTG in number operation skills (*t* (105) = 4.27, *p* ≤ 0.001) and nearly significant differences in number production skills (*t* (105) = 2.43, *p* = 0.050). Significant interactions were found between groups and times in number production skills (*F* (2,101) = 8.00, *p* < 0.001) and problem-solving skills (*F* (2,98) = 3.25, *p* = 0.043) (see Table 1 and Table 2). Simple main effects showed that scores in number production were significantly higher after the intervention in the three groups (*F*_Math GTG_ (1,99) = 10.89, *p* = 0.001; *F*_Memory GTG_ (1,104) = 10.95, *p* = 0.001; *F_CG_* (1,100) = 53.73, *p* ≤ 0.001). The change in the Math GTG (*d* = −0.46) and the Memory GTG (*d* = −0.76) in comparison to the CG showed a small and medium effect size in number production skills (see Figure 3F), respectively. Simple main effects showed that scores in problem solving were significantly higher after the intervention in both the Memory GTG (*F* (1,102) = 47.53, *p* ≤ 0.001) and CG (*F* (1,97) = 5.57, *p* = 0.020), but not in the Math GTG (*F* (1,96) = 2.50, *p* = 0.117). The change in the Math GTG in comparison to the CG did not show any effect size in problem-solving skills (*d* = −0.04). However, the change in the Memory GTG in comparison to the CG showed a small effect size in problem-solving skills (*d* = 0.41) (see Figure 3G).

### 3.5. Educative and Memory Profile of the Games

The games used in the present study were selected because the primary and most evident process that was activated while playing was memory for the Memory GTG and maths for the Math GTG. However, it might be possible that compatible processes were active in a secondary way. To test this, we analysed the games according with to an expert committee’s methodology proposed in [71] and used previously as a post-hoc analysis [72]. As can be seen in Figure 4, most of the games used in the Math GTG activated memory outcomes in addition to math skills while playing. However, most of the games used in the Memory GTG did not activate math skills while playing.

## 4. Discussion

This study assesses the effects of a cognitive intervention based on filler board games to improve STM and WM updating processes and mathematical skills in children from 8 to 10 years old. This study could have relevant implications in terms of the early promotion of children’s math learning and cognitive development through the implementation of similar training programs. Following the study of Ramani et al. [21], a group of students played memory games (general domain) whereas another group of students played mathematical games (specific domain) compared to a control group.

Firstly, we found that SES status and mathematical skills were different between those children who were included in the analysis and those who were not included. Most of the participants removed from the analysis were excluded because parents or children did not complete assessments. Ready [73] found that SES status predicts absenteeism at school. As in previous studies [74], we found that SES status correlated with mathematical skills. Hence, maybe those families that were more involved in educative aspects (i.e., to answer the questionnaires) were from higher SES levels.

Results showed that all of the groups improved their levels in many outcomes, except for verbal STM, from baseline to post assessments. Previous studies found similar results in samples of children [22]. We can explain this progress in STM, WM updating and/or mathematical skills as due to development and school instruction [75]. Another explanation could be that we repeated the same tasks a second time.

Testing the hypotheses of the present study, we found significant transfer effects of playing board games. In third grade, those children who played memory board games increased their visuospatial WM updating in comparison to the CG due to a near transfer effect. General memory training does not equally involve STM and WM updating processes. For example, all of the games that were played in the Memory GTG involved visuospatial elements but none of them involved verbal stimulus. In addition, the most trained process was visuospatial WM updating. This result could make sense considering near transfer effects of general memory training [13,19]. Unexpectedly, in third grade, the Math GTG (who played math games) revealed the greatest improvement in visuospatial STM and WM updating processes compared to the CG. Not only were STM and WM updating directly trained when playing memory games, but they were also indirectly boosted when playing mathematical games. Previous studies showed that STM and WM updating are cognitive processes underlying mathematical skills [9]. Moreover, WM and mathematical activities share common neurobiological substrates, such as frontoparietal regions [76,77]. Thus, it is possible that playing mathematical board games may increase STM and WM updating because these processes were active while playing, producing a far transfer effect. However, *Auf zack!* [67]—a game played in the specific math intervention—has some STM mechanics that could even increase the activation of the STM. In order to clarify these explanations, future studies should compare specific math trainings with [20] and without [21,22] a direct boosting of STM and WM updating. As Johann & Karbach [78] stated, a combination of cognitive training with domain-specific training could be more effective than each training alone.

Regarding mathematical skills, all of the participants increased their number operations and number ranking skills, but the Math GTG had a higher increase in comparison to the CG. As can be seen in the description of the games, math training does not equally tap the mathematical skills. All of them involved calculation skills. Our results suggest that near transfer effects are possible after specific training with mathematical tasks. Previous studies which involved a specific math training [22] and another study which involved a mixed memory and math training [20] found improvements in number operations. Related to far transfer, we found that children who played board games also improved in number ranking skills. Previous studies found that order ascending is needed in number operations [79].

In fourth grade, we found an interaction in problem solving. Although both the Memory GTG and CG increased their scores in problem solving, children who played STM/WM updating board games showed a small effect size in their increase in comparison to the CG. Previous studies suggested that problem solving and number operations are the most related math skills to memory [80]. For instance, Kuhn & Holling [22] compared the effects of a WM computerized intervention with a number sense intervention. As in our study, the WM intervention improved problem solving, whereas the number sense intervention improved number operation skills. On the one hand, previous studies suggested executive functions are needed to correctly solve problems [9,81]. Maybe board games included in the Memory GTG are more executively demanding than games included in the Math GTG. On the other hand, maybe the improvement of mathematical skills depends on the trained tasks. The mathematical board games directly trained number operations and number ranking skills, but not number production or problem solving. However, Out of Mine! [61], included in the STM/WM updating board games, has a scoring system similar to the problem-solving task. For example, players need to add or substrate points depending on their pieces and cards. Hence, adding some board games that directly trained mathematical problem solving may change the present results. Further studies are needed to elucidate this matter considering the interesting implications in school interventions.

The analysis of the cognitive and educative profile of the games can help in understanding the results. Though all of the games used in the Math Game Training Group activated math skills and memory competencies, only one game in the Memory Game Training Group activated math skills to some degree. Though some studies have found significant effects on math skills after using cognitive board games [82], other studies point out that using board games that activate more cognitive/educative processes than the most obvious ones increase the benefits of playing them [72]. Considering the results as a whole, it seems that using modern board games in the classroom with a high explicit content of maths and a high implicit content in memory is better for improving both memory functions and math skills than games that activate only memory outcomes. Hence, the previous analysis of the games can help us in future studies to better select the games and to allow us to make specific recommendations.

Finally, developmental patterns could explain the different results in third and fourth grades. Previous studies suggested that younger and older children use different strategies to solve number operations. As Raghubar et al. (2010) [83] stated in their review, although visuospatial STM and WM are general domains, these cognitive processes are more specific to early math skills than other executive processes which may be more generic in terms of supporting learning. For example, McKenzie et al. (2003) [84] suggested that younger children use a combination of verbal and visual strategies to solve calculation problems whereas older children rely more on verbal strategies. These could explain why only third grade children improve in visuospatial memory tasks. In addition, executive skills in the early primary grades may help mathematical learning and performance, but they are not necessarily associated with mathematical performance across ages or across all mathematical tasks or all domains of mathematics [80]. As Raghubar et al. [83] and de Souza-Salvador et al. [85] suggested, growth in different components of working memory and executive processes differentially predicts growth in different mathematical skills such as number operations compared to problem solving. These authors conclude that this information is needed to construct a developmental model which relates math and cognition.

### Generalizability and Limitations of the Study

First of all, the procedure of the present study could be generalizable to school settings due to teachers administering both interventions. Nonetheless, teachers were aware of the group conditions, which could influence the results obtained. For example, teachers from the CG did not apply any intervention. Following the methodology of Ramani et al. [40], future studies should use an active group that plays board games but without boosting STM, WM and mathematical skills. The present study has other drawbacks that should be considered. It is important to note that it is a quasi-experimental study. We found some baseline differences in some outcomes. For theoretical reasons, some of these variables were introduced as covariates. Future studies should carry out randomized controlled trials to increase generalizability. This is in line with the results of Sala & Gobet [17], who found that the randomization of participants and the inclusion of an active control group could reduce the significance of the results. In addition, no follow-up were performed in the current study. Future studies should include follow-ups to prove if effects persist after retiring the interventions.

Other limitations are related to interventions and outcomes. On the one hand, trainings did not equally tap basic mathematical skills. The most trained skill was number operations. However, many mathematical board games tapped number production or problem solving. In addition, some games in the specific domain intervention include memory mechanics and some games in the general domain intervention boosted mathematical skills. Thus, future studies should include board games previously analysed to apply “pure” specific and general interventions. Regarding the difficulty of the board games, Holmes et al. [86] explained the importance of adapting interventions. Although it could be more difficult in board games than in computerized training, the skill levels of the other players might be considered. In the present study, the playing groups were formed randomly. Hence, it could be possible to find higher increases in the outcomes if children with the same initial level formed the playing groups. Future studies are needed to elucidate this question.

In addition, we focused only on STM and WM updating outcomes, but not on other manipulative complex WM tasks. In future studies, we should also assess executive functions to test the hypothesis that the children who improved in problem solving improved because they activated other executive functions while were playing. As in previous studies [87], future studies should not only include neuropsychological tasks, but also behavioural rating scales to assess transfer to everyday behaviours. Additionally, in future studies it would be advisable to assess other variables associated to cognition (i.e., physical activity [88]), to mathematical skills (i.e., time spent on homework [89]) or to trainings based on games (i.e., motivation versus board games or time spent playing [25]).

## 5. Conclusions

According to our results, using board games in classroom settings could have some beneficial effects. In general, we found that mathematical and memory board games could help development in childhood in comparison to standard lessons. However, we need to take into account developmental patterns to understand the benefits of playing at school. On the one hand, we found that children between 8 and 9 years old who played in the classroom improved their visuospatial short-term memory, working memory updating, number operations and number ranking skills more than those who did these lessons without playing. On the other hand, those children between 9 and 10 years old who played memory games improved problem-solving skills, but with a small effect size. Thus, the usage of board games can be recommended as a complementary tool to usual educative practices, especially in third grade. Moreover, the intervention has great potential for being easy to incorporate in ecological settings in a cost-effective way. More randomized controlled trial studies are needed to elucidate the usage of board games as a tool in school interventions.

## Figures and Tables

**Figure 1 brainsci-14-00642-f001:**
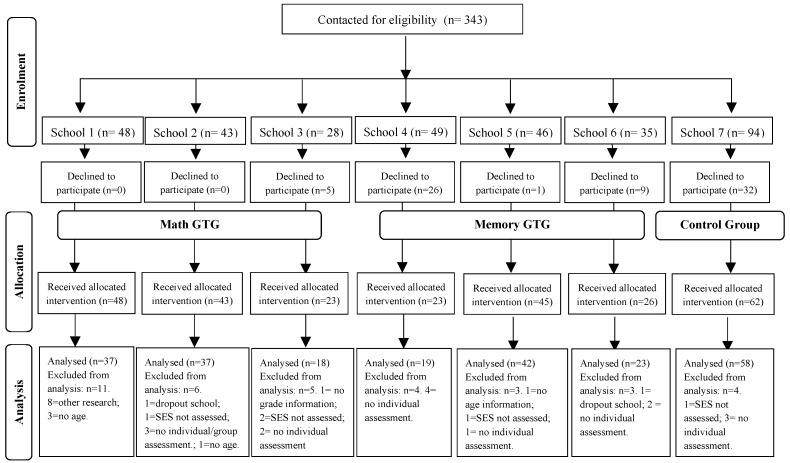
Diagram flow for the participants enrolled, allocated and analysed.

**Figure 2 brainsci-14-00642-f002:**
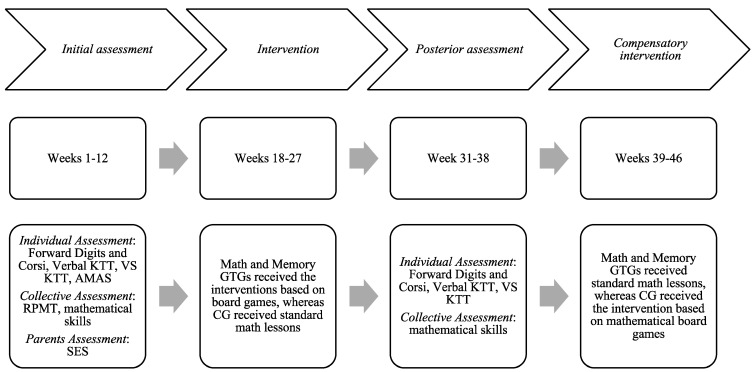
Timeline with the time ranges of the assessments and the interventions. Note. AMAS = Anxiety Math Abbreviated Scale; CG = control group; KTT = Keep Track Task; RPMT = Raven Progressive Matrices Test; SES = social status; GTG = game training group; VS = Visuospatial.

**Figure 3 brainsci-14-00642-f003:**
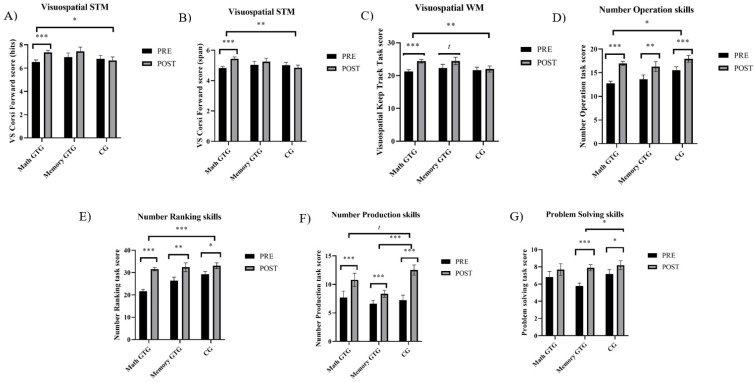
Plots from interaction results divided into memory and mathematical measures. Note. (**A**) Visuospatial STM (hits) in third grade; (**B**) visuospatial STM (span) in third grade; (**C**) visuospatial WM in third grade; (**D**) number operation skills in third grade; (**E**) number ranking skills in third grade; (**F**) number production skills in fourth grade; and (**G**) = problem-solving skills in fourth grade. * *p* < 0.05. ** *p* < 0.01. *** *p* < 0.001.

**Figure 4 brainsci-14-00642-f004:**
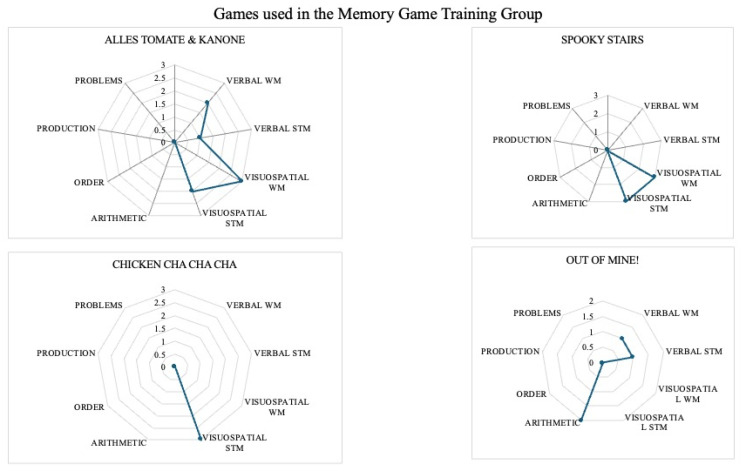
Cognitive and educative profile of the games.

**Table 1 brainsci-14-00642-t001:** Demographical/psychological characteristics and baseline outcomes for grades 3 and 4.

	Grade 3					Grade 4				
	Math GTG (*n* = 75) ^c^	Memory GTG (*n* = 18)	CG (*n* = 28)	$\chi^{2}$ (*p* Value)	ε^2^/*V*	Math GTG (*n* = 17) ^c^	Memory GTG (*n* = 66)	CG (*n* = 30)	$\chi^{2}$ (*p* Value)	ε^2^/*V*
		Demographical characteristics				Demographical characteristics				
*Age (in years) M ± SD*	8.31 ± 0.38	8.47 ± 0.36	8.46 ± 0.28	7.13 * (0.028)	0.06	9.42 ± 0.26	9.39 ± 0.40	9.39 ± 0.32	0.79 (0.673)	0.01
*Sex* ^a^				4.66 (0.097)	0.20				1.64 (0.441)	0.12
Boys, *n* (%)	46 (62%)	14 (78%)	13 (46%)			6 (35%)	34 (52%)	13 (43%)		
Girls, *n* (%)	28 (38%)	4 (22%)	15 (54%)			11 (65%)	32 (48%)	17 (57%)		
*SES Index, M ± SD*	32.80 ± 11.14	33.89 ± 9.74	35.46 ± 12.16	1.26 (0.532)	0.01	25.74 ± 11.11	30.62 ± 12.01	36.47 ± 15.01	**8.92 * (0.012)**	**0.08**
*Ethnicity* ^b^				0.64 (0.728)	0.08				7.86 (0.447)	0.20
Spanish	57 (92%)	15 (94%)	27 (96%)			10 (100%)	54 (87%)	29 (97%)		
European	5 (8%)	1 (6%)	1 (4%)			0 (0%)	5 (8%)	0 (0.0%)		
Others	0 (0%)	0 (0.0%)	0 (0.0%)			0 (0%)	3 (5%)	1 (3%)		
		Psychological characteristics				Psychological characteristics				
*Fluid reasoning, M ± SD*	29.32 ± 9.38	28.72 ± 9.83	33.43 ± 8.09	4.64 (0.098)	0.04	31.88 ± 9.82	32.56 ± 8.92	31.70 ± 9.86	0.01 (0.993)	0.00
*AMAS, M ± SD*	22.81 ± 6.43	18.67 ± 4.30	18.89 ± 6.87	**11.11 ** (0.04)**	0.09	23.59 ± 7.26	21.45 ± 7.09	18.77 ± 5.90	5.19 (0.075)	0.05
		Baseline outcome levels				Baseline outcome levels				
Verbal STM (hits), *M ± SD*	7.30 ± 1.64	6.83 ± 1.65	6.96 ± 1.84	1.88 (0.390)	0.02	7.00 ± 1.90	7.41 ± 1.75	8.20 ± 2.04	5.06 (0.080)	0.05
Verbal STM (span), *M ± SD*	5.03 ± 0.99	4.78 ± 0.94	4.96 ± 0.96	1.02 (0.600)	0.01	4.81 ± 1.11	5.18 ± 1.04	5.50 ± 1.11	4.49 (0.106)	0.04
VS STM (hits), *M ± SD*	6.49 ± 1.56	6.89 ± 1.53	6.93 ± 1.76	0.80 (0.669)	0.01	7.06 ± 1.73	6.67 ± 1.49	7.60 ± 2.04	**7.47 * (0.024)**	**0.07**
VS STM (span), *M ± SD*	4.78 ± 0.91	5.00 ± 0.77	5.11 ± 1.10	1.42 (0.492)	0.01	5.00 ± 0.89	4.97 ± 0.98	5.30 ± 1.06	3.05 (0.218)	0.03
Verbal WM, *M ± SD*	21.70 ± 4.54	21.89 ± 3.32	21.75 ± 4.28	0.15 (0.929)	0.00	21.00 ± 4.38	22.26 ± 4.18	24.47 ± 4.08	**9.47 * (0.009)**	**0.09**
VS WM *M ± SD*	20.88 ± 5.42	22.28 ± 6.14	22.46 ± 5.30	2.63 (0.268)	0.02	20.31 ± 6.76	21.59 ± 4.88	24.97 ± 4.47	**11.25 ** (0.004)**	**0.10**
Number Operations, *M ± SD*	12.52 ± 4.00	13.67 ± 3.11	16.21 ± 3.44	**17.77 *** (<0.001)**	**0.15**	16.18 ± 4.64	14.80 ± 4.47	18.30 ± 4.32	**11.44 ** (0.003)**	**0.10**
Number Ranking, *M ± SD*	21.44 ± 8.49	26.11 ± 9.59	30.04 ± 4.26	**30.17 *** (<0.001)**	**0.25**	29.65 ±9.73	29.23 ± 9.24	33.47 ± 7.61	4.75 (0.093)	0.04
Number Production, *M ± SD*	4.96 ± 3.95	5.22 ± 5.38	6.29 ± 4.66	1.53 (0.465)	0.01	7.06 ± 4.10	6.61 ± 5.14	7.67 ± 5.14	1.24 (0.538)	0.01
Problem Solving, *M ± SD*	4.35 ± 2.24	4.11 ± 2.30	7.00 ± 2.82	**19.52 *** (<0.001)**	**0.16**	6.47 ± 2.83	5.77 ± 3.04	7.37 ± 3.85	3.69 (0.158)	0.03

Note. AMAS = Anxiety Math Abbreviated Scale; CG = Control Group; GTG = Game Training Group; SES = Social Status; STM = Short-Term Memory; VS = Visuospatial; WM = Working Memory. ^a^. Sex was only registered in a subset of children in Math GTG in grade 3 (*n* = 74). ^b^. Ethnicity was only registered in a subset of children at pretest in grade 3 (*n* = 106) and in a subset of children at pretest in grade 4 (*n* = 102). ^c^. Memory Tests were only administered to a subset of children in Math GTG in grade 3 (*n* = 74) and to a subset of children in Math GTG in grade 4 (*n* = 16). * *p* < 0.05. ** *p* < 0.01. *** *p* < 0.001. ε^2^ < 0.02 = trivial; 0.02 ≤ ε^2^ < 0.13= small; 0.13 ≤ ε^2^ < 0.26 = medium; ε^2^ ≥ 0.26= large. *V* < 0.10= trivial; 0.10 ≤ *V* < 0.30= small; 0.30 ≤ *V* < 0.50= medium; *V* ≥ 0.50 = large.

**Table 2 brainsci-14-00642-t002:** Before and after intervention, memory scores (Mean ± SE), math scores (Mean ± SE) and mixed model results for grades 3 and 4.

		Grade 3 ^a^						Grade 4 ^b^					
		Math GTG (*n* = 75)	Memory GTG (*n* = 18)	CG (*n* = 28)	Time (*p*)	Group (*p*)	Timex Group (*p*)	Math GTG (*n* = 17)	Memory GTG (*n* = 66)	CG (*n* = 30)	Time (*p*)	Group (*p*)	Timex Group (*p*)
Verbal STM (hits)	Pre	7.31 ± 0.18	6.92 ± 0.37	6.88 ± 0.30	1.12 (0.292)	0.74 (0.480)	0.59 (0.553)	7.20 ± 0.50	7.43 ± 0.24	8.04 ± 0.37	0.12 (0.730)	0.46 (0.630)	0.96 (0.386)
	Post	7.30 ± 0.18	7.37 ± 0.37	6.99 ± 0.30				7.58 ± 0.50	7.68 ± 0.24	7.67 ± 0.37			
Verbal STM (span)	Pre	5.03 ± 0.11	4.84 ± 0.22	4.91 ± 0.17	1.34 (0.250)	0.71 (0.493)	0.62 (0.539)	4.92 ± 0.25	5.20 ± 0.12	5.41 ± 0.19	0.89 (0.349)	0.79 (0.458)	0.59 (0.554)
	Post	5.14 ± 0.11	5.17 ± 0.22	4.87 ± 0.17				5.17 ± 0.25	5.35 ± 0.12	5.35 ± 0.19			
VS STM (hits)	Pre	6.53 ± 0.18	6.94 ± 0.37	6.80 ± 0.30	**4.66 *** **(0.033)**	0.66 (0.517)	**3.32 *** **(0.040)**	7.06 ± 0.38	6.66 ± 0.19	7.61 ± 0.28	**3.96 *** **(0.049)**	**4.90 **** **(0.009)**	0.53 (0.590)
	Post	7.35 ± 0.18	7.44 ± 0.37	6.66 ± 0.30				7.19 ± 0.38	7.28 ± 0.19	8.04 ± 0.28			
VS STM (span)	Pre	4.82 ± 0.11	5.03 ± 0.22	5.02 ± 0.18	3.25 (0.074)	0.74 (0.481)	**5.19 **** **(0.007)**	4.99 ± 0.24	4.97 ± 0.12	5.30 ± 0.17	3.07 (0.082)	**3.55 *** **(0.032)**	0.57 (0.566)
	Post	5.44 ± 0.11	5.25 ± 0.22	4.84 ± 0.18				4.99 ± 0.24	5.33 ± 0.12	5.70 ± 0.17			
Verbal WM	Pre	22.09 ± 0.90	21.78 ± 0.44	21.42 ± 0.73	**13.30 ***** **(<0.001)**	0.21 (0.808)	0.24 (0.790)	21.32 ± 1.01	22.30 ± 0.50	24.21 ± 0.74	**9.30 *** **(0.003)**	1.97 (0.144)	1.27 (0.284)
	Post	23.93 ± 0.90	23.23 ± 0.44	23.56 ± 0.73				23.82 ± 1.01	24.03 ± 0.50	24.64 ± 0.74			
VS WM	Pre	21.23 ± 0.57	22.30 ± 1.15	21.66 ± 0.93	**13.72 ***** **(<0.001)**	0.80 (0.452)	**3.55 *** **(0.032)**	20.81 ± 1.09	21.67 ± 0.53	24.53 ± 0.80	**33.65 ***** **(<0.001)**	**5.10 **** **(0.008)**	0.54 (0.583)
	Post	24.38 ± 0.57	24.46 ± 1.15	22.01 ± 0.93				24.25 ± 1.09	24.85 ± 0.53	26.76 ± 0.80			
Number Operations	Pre	12.76 ± 0.44	13.63 ± 0.89	15.56 ± 0.72	**61.68 ***** **(<0.001)**	3.06 (0.050)	**3.73 *** **(0.027)**	16.51 ± 1.00	14.81 ± 0.50	18.06 ± 0.76	**42.20 ***** **(<0.001)**	**9.33 ***** **(<0.001)**	0.50 (0.606)
	Post	16.96 ± 0.44	16.29 ± 1.04	17.96 ± 0.72				18.63 ± 1.00	17.36 ± 0.53	21.23 ± 0.77			
Number Ranking	Pre	21.69 ± 0.77	26.39 ± 1.58	29.22 ± 1.28	**48.78 ***** **(<0.001)**	**7.13 **** **(0.001)**	**6.22 **** **(0.003)**	30.86 ± 2.00	29.35 ± 1.00	32.45 ± 1.52	**37.26 ***** **(<0.001)**	2.44 (0.0.92)	0.24 (0.789)
	Post	31.57 ± 0.78	32.41 ± 1.97	33.11 ± 1.28				35.62 ± 2.00	34.90 ± 1.08	38.90 ± 1.54			
Number Production	Pre	5.28 ± 0.53	5.25 ± 1.09	5.39 ± 0.88	**22.97 ***** **(<0.001)**	0.06 (0.943)	0.09 (0.914)	7.69 ± 1.16	6.63 ± 0.58	7.24 ± 0.88	**60.92 ***** **(<0.001)**	**3.48 *** **(0.034)**	**8.00 ***** **(<0.001)**
	Post	7.78 ± 0.54	7.30 ± 1.28	8.04 ± 0.88				10.81 ± 1.16	8.36 ± 0.61	12.53 ± 0.89			
Problem Solving	Pre	4.53 ± 0.26	4.19 ± 0.54	6.48 ± 0.43	**8.08 **** **(0.005)**	**7.36 ***** **(<0.001)**	1.71 (0.185)	6.80 ± 0.68	5.77 ± 0.34	7.17 ± 0.52	**27.40 ***** **(<0.001)**	1.13 (0.327)	**3.25 *** **(0.043)**
	Post	5.88 ± 0.27	5.10 ± 0.66	6.80 ± 0.43				7.68 ± 0.68	7.89 ± 0.36	8.18 ± 0.52			

Note. CG = Control Group; GTG = Game Training Group; SE = Standard Error; SES = Social Status; STM = Short-Term Memory; VS = Visuospatial; WM = Working Memory. ^a^. In third grade, memory tests were administered to a subset of children in Math GTG (*n* = 74). Mathematical tasks were administered to a subset of children at post-test in Math GTG (*n* = 73) and in CG (*n* = 11). ^b^. In fourth grade, memory tests were administered to a subset of children in Math GTG (*n* = 16) and Memory GTG (n_Verbal WM_ = 65). Mathematical tasks were administered to a subset of children at post-test in Memory GTG (*n* = 54) and in CG (*n* = 29). ** p* < 0.05. ** *p* < 0.01. *** *p* < 0.001.

## Data Availability

The data presented in this study are available on request from the corresponding author due to due to privacy and ethical restrictions.

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
