# Peer review of "Benefits of Playing at School: Filler Board Games Improve Visuospatial Memory and Mathematical Skills"

_brainsci, 2024, doi:10.3390/brainsci14070642_

Round 1

Reviewer 1 Report

Comments and Suggestions for Authors

The article is aimed at investigating the effect of board games for cognitive development of schoolchildren.

The topic that is developed in this article is very important and relevant from both fundamental and applied points of view. The gap of the study can be seen from this sentence within the paper: no study has been developed in school-age children training STM, Updating-WM and mathematical skills through specific and general domain interventions.

Compared with other published material, this study adds to the subject area the information about the effectiveness of a domain and specific trainings based on filler board games in a school setting in children.

I have no questions about the methodology of the study.

The references are appropriate.

I have a few comments on the article revision.

I do not think that such phrases are suitable for a scientific paper: 'The first, the second and the last authors of the study.'

Figure titles should be more informative: 'Figure 2. Timeline'.

I suggest moving games' description to the Appendix.

Comments on the Quality of English Language

Make the proofreading. Some grammar errors are in the text. For example:

'The subsample 1 consist of children who were in third grade (n=121) and the 181 subsample 2 consist of children who were in fourth grade.'

'All the analysis were ran with Jamovi 1.6.15 version'

Author Response

REVIEWER 1

Dear Reviewer,

Thank you so much for your review. It has been of great help to enhance the clarity of the manuscript. We have made some changes to the original manuscript. In red, we have highlighted the text we want to remove, and in blue the text we want to add. In addition, you can see an answer to any comment in the present file. Please, tell us whether you agree with the changes or we need to modify anything else.

The article is aimed at investigating the effect of board games for cognitive development of schoolchildren.

The topic that is developed in this article is very important and relevant from both fundamental and applied points of view. The gap of the study can be seen from this sentence within the paper: no study has been developed in school-age children training STM, Updating-WM and mathematical skills through specific and general domain interventions.

Compared with other published material, this study adds to the subject area the information about the effectiveness of a domain and specific trainings based on filler board games in a school setting in children.

I have no questions about the methodology of the study.

The references are appropriate.

I have a few comments on the article revision.

Thank you for the introduction of this review. We are very appreciative of your opinion of the article. We answered each of your comments below.

I do not think that such phrases are suitable for a scientific paper: 'The first, the second and the last authors of the study.'

Thank you for your comment. We changed these phrases for the term “psychologists and senior research psychologist” (see Page 9, Lines 323 and 324).

Figure titles should be more informative: 'Figure 2. Timeline'.

Thank you for your comment. We added some information to the Figure Titles (See Figure 1 in Page 7, Figure 2 in Page 9, and Figure 3 in Page 15).

I suggest moving games' description to the Appendix.

Thank you for this comment. We considered it and we moved the description of the games to Appendix (See Page 10, Line 342 and Page 11, Line 354). We also changed Table 3 for Table 2 in all the manuscript.

***However, it is necessary to change Table 3 for Table 2 in Page 17.

Reviewer 2 Report

Comments and Suggestions for Authors

This is an interesting paper investigating the effect of filler board games on visuospatial memory and mathematical skills. The paper is well-written with very good sample size. I have a few comments to improve the manuscript further:

1. The identification of gaps in the literature is clear and well-supported. The paper should further explore why these gaps exist and how this study specifically addresses them.

2. While the study aims are clear, the introduction would benefit from a brief discussion on the methodological approach, particularly why filler board games are expected to be effective. The discussion about near and far transfer in the introduction is good, but the authors should link the discussion with the current study.

3. The rationale for the sample size calculation is clear and well-founded on previous research. However, it would be helpful to explain why a 50% risk of possible losses was considered

4. The inclusion and exclusion criteria are comprehensive and appropriate.

5. The selection of covariates is appropriate and based on relevant literature

6. The use of appropriate statistical tests for baseline differences is commendable. The detailed explanation of post-hoc tests and effect size calculations is thorough.

7. The acknowledgment of differences in SES status and mathematical skills between included and excluded participants is important. Discussing how these differences might have influenced the results would be useful

Author Response

REVIEWER 2

Dear Reviewer,

Thank you so much for your review. It has been of great help to enhance the clarity of the manuscript. We have made some changes to the original manuscript. In red, we have highlighted the text we want to remove, and in blue the text we want to add. In addition, you can see an answer to any comment in the present file. Please, tell us whether you agree with the changes or we need to modify anything else.

This is an interesting paper investigating the effect of filler board games on visuospatial memory and mathematical skills. The paper is well-written with very good sample size. I have a few comments to improve the manuscript further:

  1. The identification of gaps in the literature is clear and well-supported. The paper should further explore why these gaps exist and how this study specifically addresses them.

Thank you for your comment. Although these board games are commercialized and can be used in school settings, there is a lack of scientific evidence in the literature. We could explain this gap because many of the interventions were based on computerized interventions emphasizing this statement in the introduction (See Page 3, Line 14). Hence, these could explain why there is a lack of the use of other methodologies, such as board games.

  1. While the study aims are clear, the introduction would benefit from a brief discussion on the methodological approach, particularly why filler board games are expected to be effective. The discussion about near and far transfer in the introduction is good, but the authors should link the discussion with the current study.

Thank you for your comment. We emphasize in the introduction the characteristics of these games and why could be a good option to be included in training (Page 2, Lines 92-93, and Page 3, Lines 112-113).

We also included in the discussion section the rationale for transfer effects to explain our results, following the structure from the introduction (Page 19, Line 592 and Line 607).

  1. The rationale for the sample size calculation is clear and well-founded on previous research. However, it would be helpful to explain why a 50% risk of possible losses was considered

Thank you for this comment. We calculated the sample size calculation from a previous study (Estrada-Plana et al., 2019). From this study, the people contacted were 53 families, and finally were included 27. Hence, we considered that maybe half of the families recruited would be lost from the final data. To be better explained, we added this information to the manuscript (See Page 4, Lines 157-158).

  1. The inclusion and exclusion criteria are comprehensive and appropriate.

Thank you for this comment.

  1. The selection of covariates is appropriate and based on relevant literature

Thank you for your comment.

  1. The use of appropriate statistical tests for baseline differences is commendable. The detailed explanation of post-hoc tests and effect size calculations is thorough.

Thank you for this comment. It is very encouraging.

  1. The acknowledgment of differences in SES status and mathematical skills between included and excluded participants is important. Discussing how these differences might have influenced the results would be useful

Thank you for your comment. In the discussion section, we explained that we found these differences, but it is true that we did not discuss this finding. Hence, we added a rational explanation of this finding following our results (See Page 19, Lines 582-583).

Reviewer 3 Report

Comments and Suggestions for Authors

1.       “The games used in the present studye” – typo.

2.       It is not clear why two groups of different grades were chosen for the experimentation? Where is a reason to use two groups of different ages? One group played mathematical board games and another group played memory board games. Why not the same type of the games? Can be the results of different type games contrasted and recommended one type of games? The effects of these games are different, since “most of the games used in the Maths GTG activated memory outcomes in addition to maths skills while playing. However, most of the games used in the Memory GTG did not activated maths skills while playing.

3.       There is present one but rather shy conclusion “Considering the results in a whole, it seems that using modern board games in the classroom with a high explicit content of maths is better for improving both executive functions and math skills than games that activates only memory outcomes.”. However, the results cannot be generalized, since “developmental patterns could explain the different results in third and fourth grades.”. Such a conclusion undermines the obtained result, and it does not allow to make a generic recommendation.

4.       In the abstract and throughout the manuscript, the board games were divided in two groups, mathematical and memory. However, in the hypotheses, we find a new term “filler board games” that is a generic one. “We hypothesized that: i) those children who were trained at playing filler board games would improve their STM and Updating-WM abilities after the intervention greater than the control group; ii) those children who were trained at playing filler board games would improve their mathematical skills after the intervention greater than the control group.”. The generic conclusion was expected; however, it was not received.

5.       Conclusion is as follows: the research is quite messy, since two groups of different ages were playing two different types of games. My recommendation is not to contrast these two groups and to make two different manuscripts for each group separately.

Comments on the Quality of English Language

Moderate editing of English language is required

Author Response

REVIEWER 3

Dear Reviewer,

Thank you so much for your review. It has been of great help to enhance the clarity of the manuscript. We have made some changes to the original manuscript. In red, we have highlighted the text we want to remove, and in blue the text we want to add. In addition, you can see an answer to any comment in the present file. Please, tell us whether you agree with the changes or we need to modify anything else.

  1. “The games used in the present studye” – typo.

Thank you for alerting us of this mistake. We corrected it (See Page 18, line 529).

  1. It is not clear why two groups of different grades were chosen for the experimentation? Where is a reason to use two groups of different ages? One group played mathematical board games and another group played memory board games. Why not the same type of the games? Can be the results of different type games contrasted and recommended one type of games? The effects of these games are different, since “most of the games used in the Maths GTG activated memory outcomes in addition to maths skills while playing. However, most of the games used in the Memory GTG did not activated maths skills while playing.”

Thanks for the present comment. We will try to explain it deeper. We differentiated groups into different grades because the levels of development of math skills in both grades are very different. In Spain, in third grade, students begin to deepen in numerical concepts and begin to perform additions and subtractions. In fourth grade, students are supposed to master these mathematical operations. We have added a sentence to clarify it (See Page 4, lines 186-188).

Regarding the question about the games, we tried to explain it in the introduction. According to Executive Function (EF) models, students could improve in maths because of a far transfer effect. If EF is enhanced, these benefits could be transferred to math skills. But also, on the contrary, it is likely that with increasing math skills, EF could increase. So, having children playing memory games while other children played math games made sense. By evaluating memory and maths, we try to see which type of games lead to the best increases in memory and math outcomes.  As the Reviewer points out, we can say that, at least in Third Grade, it seems advisable to use math games to increase basic math skills and also memory. In Fourth Grade, it seems advisable to use memory games because they helped increase complex math skills (problem-solving). However, future RCTs are needed to reinforce these results (See Page 21, lines 723-724).

  1. There is present one but rather shy conclusion “Considering the results in a whole, it seems that using modern board games in the classroom with a high explicit content of maths is better for improving both executive functions and math skills than games that activates only memory outcomes.”. However, the results cannot be generalized, since “developmental patterns could explain the different results in third and fourth grades.”. Such a conclusion undermines the obtained result, and it does not allow to make a generic recommendation.

Thank you for this comment. Indeed, this general conclusion cannot allow us to make a generic recommendation. Following the statements of this paragraph (Page 20, Lines 651-655), we highlight the importance of assessing implicit and explicit cognitive and educative processes used while we play board games. This analysis can help us in future studies to better select the games and to allow us to make generic recommendations.

  1. In the abstract and throughout the manuscript, the board games were divided in two groups, mathematical and memory. However, in the hypotheses, we find a new term “filler board games” that is a generic one. “We hypothesized that: i) those children who were trained at playing filler board games would improve their STM and Updating-WM abilities after the intervention greater than the control group; ii) those children who were trained at playing filler board games would improve their mathematical skills after the intervention greater than the control group.”. The generic conclusion was expected; however, it was not received.

Thank you for this comment. As we explained in the introduction, filler board games are those that are short, with a few rules and to be played with other people. Hence, all the games used in the interventions (both in the memory and the mathematical games) are filler board games (for a detailed description, See Page 3, Lines 109-110). However, it is true that in the hypothesis it seems that we were not considering differences between the games from the interventions. Therefore, we changed the hypotheses to fit with the background and the objective of the study (See Page 3, Lines 145-142).

  1. Conclusion is as follows: the research is quite messy, since two groups of different ages were playing two different types of games. My recommendation is not to contrast these two groups and to make two different manuscripts for each group separately.

This comment is important for us because we want to improve the manuscript to be better understandable. As we explained in comment 2 (See above), it is important to consider developmental differences that can explain the results. Previous studies also divided the changes on the outcomes considering different age ranges (for instance, Vita-Barrull, N., Estrada-Plana, V., March-Llanes, J., Sotoca-Orgaz, P., Guzmán, N., Ayesa, R., & Moya-Higueras, J. (2024). Do you play in class? Board games to promote cognitive and educational development in primary school: A cluster randomized controlled trial. Learning and Instruction, 93, 101946). In addition, this study followed a non-randomized methodology (each school was allocated to each intervention). Therefore, to better control developmental differences to age, we consider that was better to show the results by dividing the sample in third and fourth grade. As we consider the limitations of the study (Page 21, Line 684), future randomized studies are needed to avoid possible differences in the baseline due to the age of the participants or other characteristics.

Round 2

Reviewer 2 Report

Comments and Suggestions for Authors

The authors have addressed my comments well. I appreciate all their efforts

Reviewer 3 Report

Comments and Suggestions for Authors

Thank you for the revision. However, your chosen way of presentation of the research results does not allow to make a generic recommendation.

Comments on the Quality of English Language

Moderate editing of English language is required